# Financial sector development, external debt, and Turkey's renewable energy consumption

**Majdi Saleem Jabari[1], Mehmet Aga[1], Ahmed Samour[2]\***

**1** Accounting and Finance Department, Cyprus International University, Nicosia, North Cyprus, **2** Banking and Finance Department, Near East University, Nicosia, North Cyprus

\* ahmad.samour@neu.edu.tr

## Abstract

The primary aim of this paper is to provide fresh evidence by testing the linkage between renewable energy consumption, financial development, and external debts in Turkey, using the Bootstrap ARDL test (McNown et al. 2018). The Bootstrap ARDL test is desired over traditional co-integration tests due to its ability to predict when resolving power and size limitation issues, and its corresponding features, which have not been addressed by traditional co-integration tests. The ARDL testing model is employed to investigate the coefficients amongst the selected variables. The findings from the ARDL test illustrate that there is a positive linkage between renewable consumption and Turkey's financial development. Furthermore, the outcomes illustrate that the coefficient of external debt is negative and significant. The results indicate that policymakers in Turkey must use the growth of the financial sector to minimize environmental degradation by promoting investment in energy and production through renewable energy sources. Furthermore, the research suggested that Turkey's policy-makers should reformulate the external debt policy to reduce the negative influence of external debt on sustainable energy development. This could potentially be achieved by removing any restrictions on international capital flow or barriers on foreign capital and foreign investment. Hence, the findings of this paper provide valuable conclusions and recommendations for Turkey heading to sustainable and green financial sector.

## 1. Introduction

Turkey is an emerging country and Turkey's energy consumption has rapidly grown especially over the last decades. For instance, Turkey's oil consumption has risen from around 313,000 to 762,500 barrels each day over the period from 1980 to 2016. Whereas, the production of oil in Turkey rose from 41,000 barrels per/day to around 57,500 barrels per/day over the period from 1980–2016 (Fig 1). As a result of increasing oil consumption, greenhouse gas emissions (GHG) have increased from 1.72 metric tons in 1980 to 4.48 metric tons in 2014. On other hand, consumption of renewable energy as percentage of total energy consumption has decreased from 26% in 1985 to 12% in 2018 (Fig 2). In this sense, Turkey should find more sources for the energy supply formula by increasing renewable energy sources [1]. However, renewable energy resources are simply renewed and produced. Furthermore, these sources diffuse fewer pollutants to nature [2]. The main renewable energy sources in Turkey are biomass,

**Data Availability Statement:** Data are available from: https://data.worldbank.org/indicator/FS.AST.DOMS.GD.ZS https://data.worldbank.org/topic/20 https://data.worldbank.org/indicator/NY.GDP.

MKTP.CD?locations=1W https://data.worldbank.org/indicator/EG.FEC.RNEW.ZS.

**Funding:** The funders had no role in study design, data collection and analysis, decision to publish, or preparation of the manuscript

**Competing interests:** The authors have declared that no competing interests exist.

**Fig 1. Oil production and consumption for 1980 to 2016 period in Turkey.** Source: world meters.

wind, hydroelectric, solar, waves, and, geothermal. In this sense, Turkey has strong potential for growth of geothermal energy, including for power generation, and heating [3]. Furthermore, Turkey has high solar energy potential. According to the Ministry of Energy and Natural Resources of Turkey, the annual total sunshine duration is around 7.2 hours, and the annual total solar energy is 1,527 kilowatt-hours per square meter ($kWh/m^2$). Turkey is in the first top 10 countries in terms of producing wind energy. Particularly Çanakkale and İzmir have very huge wind energy potential. Besides hydroelectricity potential of Turkey is equal to 16% of the hydroelectricity potential of Europe in terms of the economic sector. Moreover, Aegean, Mediterranean, and the Black Sea have high wave energy potential due to the geographic situations of Turkey [4]. However, Turkey does not have adequate technology to take advantage of wave energy; for instance, new systems should be developed for various wave sizes.

Many empirical studies have explored the linkage between economic growth, and renewable consumption of energy; however, few studies tested the possible influence of external debt on their frameworks. However, the current paper aims to provide new perspectives to the

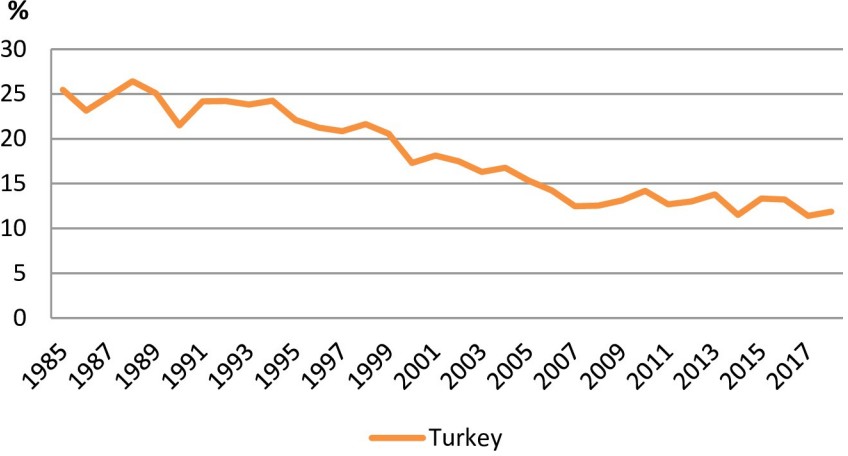

**Fig 2. Renewable energy consumption for the period from 1985 to 2018 in Turkey.** Source: world meters.

empirical literature by exploring the linkage among real income, external debt, financial sector development, and the level of renewable energy consumption in Turkey, using the developed technique of bootstrap-ARDL model as introduced by [5]. In this context, the present study aims to present novel empirical evidence by testing the linkage among external debt, financial development, and Turkey's renewable energy consumption using the novel techniques of bootstrap (ARDL) testing (McNown et al. 2018) [5].

However, the present study provides two main contributions to the current literature. First, to the best of our research knowledge, no empirical study examined the linkage among external debt and Turkey's renewable energy consumption. Second, the current study provides robustness analysis by examining the linkage amongst the selected variables employing the novel techniques of bootstrap ARDL, this is desired over traditional co-integration tests due to its ability to predict when resolving power and size limitation issues, and its corresponding features, which have not been addressed by traditional co-integration tests. Furthermore, the current study aims to present valuable recommendations for Turkey's Policymakers to achieve the sustainable environment.

The development of the financial systems is a crucial concern for Turkey, which strives to be one of the top 10 economies in the world. Environmental pollution is the main barrier preventing this development in Turkey [6]. Hence, policymakers need to devote more attention to the consequences of economic policies on CO2 emissions. In essence, the financial sector makes decisions about (whom to lend) and (what to invest). Therefore, their decisions affect business practices and investment [7]. When the financial sector provides funds for the investors, this will lead to boosting total factor productivity, capital accumulations, and promoting total factor productivity, thus, it will lead to an increase in the real income, and energy consumption [8, 9]. By taking the influence of financial sector development on the consumption of renewable energy, the current research can test the influence of financial sector policies on the consumption of renewable energy; in fact, the financial sector has a crucial role in attaining economic and financial stability through credit that provided to the markets. The financial sector should use this credit to support the investment in renewable energy sources [1].

On others hand, the external debt in Turkey increased significantly over the last decades from 19 billion US dollars in 1980 to 431 billion US Dollars in 2020 (Fig 3). Furthermore, the external debt in Turkey as a share of GDP increased from 25% to 50% in 2016. The enormous increment in external debt in Turkey may be attributed to the decrease in foreign capital

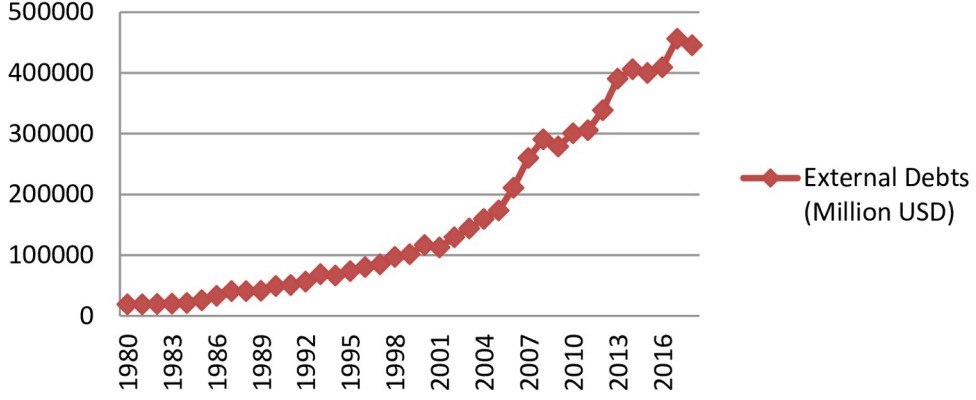

**Fig 3. External debt in Turkey for the period from 1980 to 2016 in Turkey.** Source: world Bank.

inflow, and government revenues. However, with limited foreign exchange reserves in Turkey, local currency to US Dollar tumbled as much as 300% over the last 5 years. While the majority of Turkey's external debt is composed in the US dollars, enormous increment in the level of external debt. All these factors increase the significant influence of external debt on Turkey's economic performance. According to the literature, external debt in is evaluated as a negative impact on economic performance [10–15].

However, testing the linkage among external debts, and energy consumption is still limited. In this sense, this research aims to inform policymakers about the linkage among the external debt and renewable energy consumption. The study suggests that there are several ways in which external debt may affect the consumption of energy. The first way is through the negative linkage between debt and real income. Theoretically, the debt overhang theory demonstrated that there is a negative linkage among debt and real income. The theory explained that the local and foreign investments will be decreased when the total debt exceeds the repayment ability of the country, which will subsequently cause the level of productivity and income to decrease [16]. However, the external debt affects negatively the financial strength of investment, which in turn will affect negatively the level of energy consumption. The second way is through a spillover influence of external debt on the fiscal policies. An example, an increase in the levels of debts may lead to adopting policies such as (increase the tax rates) to meet the country obligations. In this sense, the high tax rates can discourage the level of saving, investment, and innovation, which in turn will affect negatively energy investment and consumption. The third way is through spillover influence of external debt on borrowing costs for the individual and markets. An increase in the borrowing cost will subsequently cause the investment to decrease. Subsequently, it will cause renewable energy consumption to decline [17].

The structure of this paper as follows: the second section of this research is a review of literature; the third and fourth sections show empirical model, research methodology, and the research findings, the last section presents the conclusion of the current study.

## 2. Review of literature

### 2.1 Real income and renewable energy consumption

Many research considered the effect of real income on energy consumption. The empirical research on the relationship among consumption of energy and real income could be shown into four hypotheses, namely; (i) the growth hypothesis (GH), if there is a unidirectional causal linkage from consumption of energy to real income, the GH will be accepted [18]. (ii) The conservation hypothesis (CH), if there is a unidirectional causal linkage from real income to consumption of energy, the CH will be accepted [19]. (ii) The feedback hypothesis (FH), if there is a bi-directional causal linkage among real income and consumption of energy, the FH will be accepted [20]. (iv) The neutrality hypothesis (NH), if there is no causal linkage amongst real income and consumption of energy, the NH will be accepted [18].

However, the effect of real income on the consumption of energy in recent years has drawn extensive attention. In this respect; [19] used the FMOLS model; the findings showed that a bidirectional association exist in 80 selected countries over the tested period 1990–2007. Using the ARDL model [20], found that there is a positive connection between real income and consumption of energy in the BRICS members for the period 1971–2010. [21] utilized the Granger causality model and explored the effect of real income on consumption of energy in G7 countries over the period 1960–2010. The outcomes confirmed that the feedback hypothesis holds in Japan. The conservation hypothesis is relevant in Italy, the growth hypothesis is relevant in Canada, and the neutrality hypothesis is relevant in France, the United States, and the UK. In Turkey, [22] utilized the Granger causality test, and discovered a significant linkage between

real income and consumption of renewable energy for the period 1970–2006, the findings showed the existence of the feedback hypothesis in Turkey. [23] confirmed that the feedback hypothesis (FH) is relevant in Turkey over the period 1992–2012. [24] indicated that Turkey has a large population and high energy consumption rates with high expectations for high economic growth. Moreover, [25], and [26] employed the causality test. The results show that there is association between Turkey's energy consumption and real income in the 1985–2016 and 1988–2018 periods respectively.

## 2.2 Financial sector development and renewable energy consumption

Many empirical studies considered the influence of the development of the financial sector on real income and energy consumption. For instance [27–29], tested the impact of the development of the financial sector on real income. In this context, these studies suggested that an increase in investment and credit to the markets led to increase the real income. However, the growth of the financial sector is one of the central parts of financial sector development; in existing literature, some empirical studies that tested the linkage among financial sector development, and renewable energy consumption. In this sense: [28] revealed a positive association between the growth of the financial sector and renewable energy consumption in Turkey. [30] tested the linkage among financial sector development and consumption of renewable energy in India, over the period 1971–2015. Using the ordinary least squares test, the findings confirmed that there is positive link among financial development, and the level of renewable energy consumption in India [31]. Showed that there is a causal linkage between insurance sector development and Turkey's renewable energy consumption. [32] used FMOLS test, and confirmed a positive association between financial development and consumption of renewable energy in BRICS countries. [33] tested the impact of financial development on consumption of renewable energy in 28 European countries, over the period 1990–2015. Using the fixed-effect panel model, the findings confirmed that financial development has a positive influence on the level of renewable energy consumption. [34] tested the impact of financial development on the consumption of renewable energy in China, over the tested period 1990–2015. Using the ARDL testing model, the findings confirmed that financial development affects positively the level of renewable energy consumption in long run. However, the findings of these papers indicate that financial growth plays an essential part in strengthening and enhancing the financial system's economic performance, which can affect economic activity and energy consumption.

## 2.3 External debt and renewable energy consumption

External debt is a component of the total government debt that a country owed to foreign lenders. The prime reason for a country borrowing from foreign creditors is that the estimated expenditure exceeds the expected revenues of the country. Theoretically, the debt overhang theory showed that there is a negative linkage between debt and real income. The theory explained that the local and foreign investments will be decreased when the total debt exceeds the repayment ability of the country, which will subsequently cause the level of productivity and income to decrease [16]. Many empirical studies tested the linkage among external debts and real income. For instance [14, 35, 36], found empirical evidence to support the negative linkage among external debts and real income. In this sense [36], tested the influence of public debt on the real income in 12 European countries over the period from the 1970–2008. The outcomes illustrated that an increase in the level of debt will affect negatively the real income. [37] used the ARDL model, and indicated that there is an inverse linkages among debt and real income in Malaysia over the period from 1991 to 2009. [16] found that there is a negative

impact of external debt on the economic growth in the middle-income countries over the tested period from 2002 to 2016. [35] tested the association between the debt and real in income in Turkey. The outcomes confirmed a negative linkage among the real income and debt in Turkey over the period from 2003–2017. However, limited papers have explored the linkage among external debt, energy consumption and environmental degradation. [38] used the ARDL model, and tested the influence of external debt on levels of carbon emissions in Turkey over the tested period from 1960 to 2013. The findings showed that there is important linkage among $CO_2$ emissions, energy, real income and external debt in Turkey. [39] employed the ARDL technique to explore the effect of public debt on the level of CO2 emissions in China over the tested period from 1978 to 2014. The findings showed a positive and significant influence of external debt on the level of environmental pollution in China. [34] tested the linkage among public debt on the consumption of renewable energy in BRICS countries, over the examined period 1990–2016. Using the Panel testing model, the findings confirmed that public debt affects negatively the level of renewable energy consumption in BRICS countries. [40] tested the linkage among public debt and consumption of renewable energy in 20 selected emerging countries, over the period 1990–2016. Using the Granger causality test, the findings confirmed that there is a causal linkage among the level of debt and renewable energy consumption. Table 1 shows the summary of Literature review.

## 3. Empirical model

The prime aim of this research is to explore the linkage among real income, financial sector development, external debt and renewable energy consumption in Turkey. The theoretical linkage between real income, financial sector development, and external debt and renewable energy consumption in Turkey is expressed as follows:

$$\ln RE^C_{it} = \beta_0 + \Upsilon_1 \ln R^I_{it} + \Upsilon_2 \ln FS^D_{it} + \Upsilon_3 \ln E^D_{it} + \varepsilon_t \tag{1}$$

**Table 1. Summary of literature of review.**

| Authors | Method | Year | Country | Result |
|---------|--------|------|---------|--------|
| Apergis and Payne (2012) | FMOLS | 1990–2007 | 80 Countries | $R^I(+)RE^C$ |
| Erdal et al., (2008) | Granger Causality | 1970–2006 | Turkey | $R^I \rightarrow \leftarrow RE^C$ |
| Sentürk and Sataf (2015) | VECM | 1971–2006 | Turkey | $R^I = RE^C$ |
| Sebri and Salha (2014) | ARDL | 1971–2010 | BRICS | $R^I(+)RE^C$ |
| Ajmi et al., (2015) | Granger Causality | 1960–2010 | G7 Countries | $R^I \rightarrow \leftarrow RE^C$ |
| Samour and Pata (2022) | Granger Causality | 1985–2016 | Turkey | $R^I \rightarrow \leftarrow RE^C$ |
| Ocal and Aslan (2013) [41] | Toda Yamato test | 1990–2010 | Turkey | $R^I \# RE^C$ |
| Beck et al., (2000) [42] | GMM | 1960–1995 | 93 Countries | $FSD^I(+) RE^C$ |
| Aslan et al., (2014) | FMOLS | 1980–2011 | 7 Middle East Countries | $FSD^I(+) RE^C$ |
| Haseeb et al., (2018) | FMOLS | 1995–2014 | BRICS countries | $FSD^I(+) RE^C$ |
| Eren et al., (2019) | DOLS | 1971–2015 | India | $FSD^I(+) RE^C$ |
| Anton and Nucu (2020) | Fixed effect model | 1990–2015 | 28 European Countries | $FSD^I(+) RE^C$ |
| Wang et al., (2021) | ARDL | 1990–2015 | China | $FSD^I(+) RE^C$ |
| Hashemizadeh et al., (2021) | Granger causality | 1990–2016 | 20 emerging countries | $E^D \rightarrow \leftarrow E^C$ |
| Wang et al., (2021) | Panel model | 1990–2016 | BRICS countries | $E^D(-) E^C$ |

$E^D$ is external debt, $RE^C$ is renewable energy consumption, $FSD^I$ is financial sector development, + means positive relation, − means negatvie relation, $\rightarrow \leftarrow$ means bidirectional causal linkage, # means no relation.

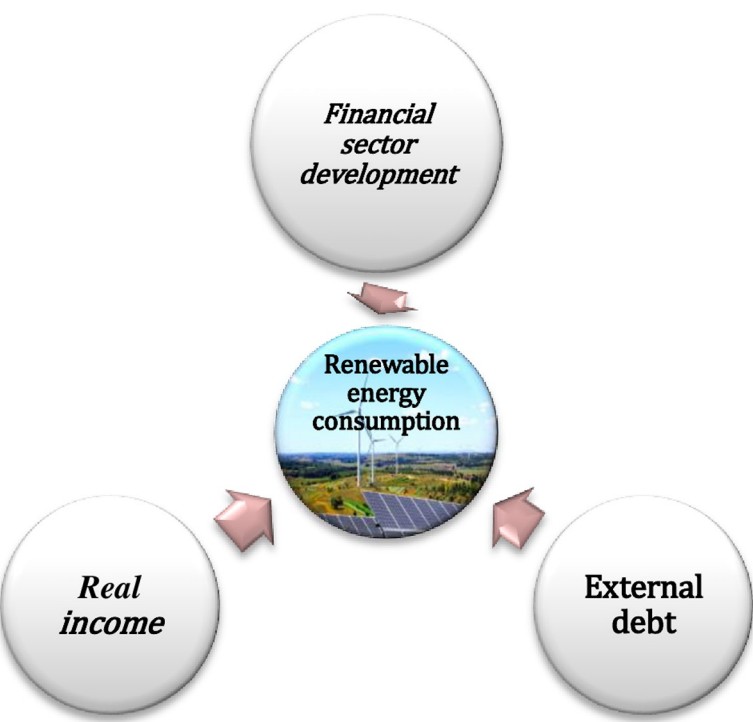

**Fig 4. Shows the linkage among the variables.**

ln is logarithm of tested variables. $RE^C$ Represents Turkey's renewable energy consumption, $R^I_{it}$ is Turkey's real income (2010 = 100) in US\$ per capita, $FS^D$ is credits provided by the Turkish banks to the (private-sector) as a share of GDP. $E^D$ is external debt stocks, total (US\$). Fig 4 shows the linkage among the variables. The data of this research is yearly data covering the period from 1980–2016. Table 2 shows a description of the tested variables and data sources.

### 3.1 Stationary and co-integration tests

The classical unit roots tests do not include any date of a structural break(D-SB); To overcome this problem, This research employs (Perron and Vogelsang 1992 [43]), and Zivot-Andrews (2002) [44] unit roots tests with endogenously determined dates of structural break (D-SB). The study uses the novel technique of ARDL as updated by [5] McNown et al. (2018) to explore the linkage between renewable energy consumption, real income, external debt, and financial sector development in Turkey. This technique contains additional of ($t-test\ t_{dependent}$) or ($F-test\ F_{independent}$) on the coefficients of lagged-independent tested variables. However, the null hypothesis of $t_{dependent}$ test is: $\partial 1 = 0$. The alternative hypothesis of $t_{dependent}$ test is: $\partial_1 \neq 0$.

**Table 2. Description of investigated variables.**

| Variable | Description |
|---|---|
| $R^I_{it}$ | GDP (2010 = 100) in US\$ per-capita |
| $RE^C_{it}$ | Renewable energy consumption as share of total of energy consumption |
| $FS^D$ | Credits which provided by the banking sector to the (private-sector) as a (%) of GDP |
| $\ln E^D$ | External debt stocks, total (US\$) |

N, source of the data World- Bank (WB).

While the null hypothesis of $F_{independent}$ test is: $H_0$: $\partial_2 = \partial_3 = \partial_4 = 0$. The null hypothesis of $F_{independent}$ test is: $H_1$:: $\partial_2 \neq \partial_3 \neq \partial_4 \neq 0$.

The bootstrap ARDL critical values (CV) include the features of the combination integration of each time-series employing the ARDL bootstrap procedures. These steps will lead to overcoming the instability issue in the findings of traditional co-integration tests, as well as it will lead to provide a better result than other tests of co-integration. In this sense, the CV of ARDL bound test allows only for (one) tested variable to be endogenous, whereas in the updated ARDL test: the critical values allow for the endogeneity of selected explored variables [31]. Also, this technique is advisable for time-series that include more than (one) examined tested variable [45]. The co-integration amongst real income, financial sector development, and external debt on renewable energy consumption in Turkey will be determined if the values of $F-_{Pesaran}$, $t-_{dependent}$, $F-_{independent}$ higher than the CV bootstrap technique [5] (McNown et al. 2018). The ARDL testing approach equation is as follows:

$$\Delta Ln\, RE^c_{\,t} = \beta_0 + \sum_{i=1}^{n}y_1\Delta lnRE^C_{\,t-j} + \sum_{i=1}^{n}y_2\Delta lnR^I_{\,t-j} + \sum_{i=1}^{n}y_3\Delta lnFS^D_{\,t-j} + \sum_{i=1}^{n}y_4\Delta lnE^D_{\,t-j}$$
$$+ \sigma_1 lnRE^c_{\,t-1} + \sigma_2 lnR^I_{\,t-1} + \sigma_3 lnFS^D_{\,t-1} + \sigma_4 lnE^D_{\,t-1} + +\omega ECT_{t-1} + \varepsilon_{1t} \qquad (2)$$

where $\varepsilon_{1t}$ symbolizes the error-term. $\Delta$ is the operator of the first difference. $ln\, RE^C_{\,t}$, $lnR^I_{\,t}$, $lnFS^D_{\,t}$, $lnE^D_{\,t}$ are variables of the study, n symbolizes the optimal of lags, which is selected bas on Akaike information criterion(AIC), the error correction model (ECM) is estimated by employing the Eq (2) $ECT_{t-1}$ is the one period lagged $EC^{term}$. It reflects the velocity of adjustment amongst the selected variable. To affirm the outcomes of the ARDL testing approach. The current research employed the following diagnostic tests the normality test ($D^{t1}$), the Breush-Pagan Godfrey heteroscedasticity ($D^{t2}$), the ARCH test ($D^{t3}$), B-Godfrey serial correlation test ($D^{t4}$), Ramsey-Reset test ($D^{t5}$), the multicollinearity test $D^{t6}$. $D^{t1}$ is utilized to affirm the normal distribution of the tested model. $D^{t2}$ is utilized to affirm that there is that no problem of heteroscedasticity exists in the tested model. $D^{t3}$ and $D^{t4}$ are utilized to affirm that there is no autocorrelation in the study model. $D^{t5}$ is utilized to affirm that the tested model is stable. The multicollinearity test $D^{t6}$ used to affirm that there is no multicollinearity. Furthermore, Granger causality method is utilized to explore the causal interaction amongst the investigated variables. In this test, ($EC^{Term}$) defines the short-term level variations of the variables studied from the long-term equilibrium level. The $ECM$ formulated in equations (3 to 6):

$$\Delta Ln\, RE^C_{\,t} = \beta_0 + \sum_{i=1}^{n}y_1\Delta lnRE^C_{\,t-j} + \sum_{i=1}^{n}y_2\Delta lnR^I_{\,t-j} + \sum_{i=1}^{n}y_3\Delta lnFS^D_{\,t-j} + \sum_{i=1}^{n}y_4\Delta lnE^D_{\,t-j}$$
$$+ \omega ECT_{t-1} + \varepsilon_{1t} \qquad (3)$$

$$\Delta Ln\, R^I_{\,t} = \beta_0 + \sum_{i=1}^{n}y_1\Delta lnR^I_{\,t-j} + \sum_{i=1}^{n}y_2\Delta lnRE^C_{\,t-j} + \sum_{i=1}^{n}y_3\Delta lnFS^D_{\,t-j} + \sum_{i=1}^{n}y_4\Delta lnE^D_{\,t-j}$$
$$+ \omega ECT_{t-1} + \varepsilon_{1t} \qquad (4)$$

$$\Delta Ln\, FS^D_{\,t} = \beta_0 + \sum_{i=1}^{n}y_1\Delta lnFS^D_{\,t-j} + \sum_{i=1}^{n}y_2\Delta lnRE^C_{\,t-j} + \sum_{i=1}^{n}y_3\Delta lnR^I_{\,t-j} + \sum_{i=1}^{n}y_4\Delta lnE^D_{\,t-j}$$
$$+ \omega ECT_{t-1} + \varepsilon_{1t} \qquad (5)$$

**Fig 5. Shows the methodology structure of this research.**

$$\Delta \text{Ln } E^D_t = \beta_0 + \sum_{i=1}^{n} y_1 \Delta \ln E^D_{t-j} + \sum_{i=1}^{n} y_2 \Delta \ln RE^C_{t-j} + \sum_{i=1}^{n} y_3 \Delta \ln R^I_{t-j} + \sum_{i=1}^{n} y_4 \Delta \ln FS^D_{t-j}$$
$$+ \omega ECT_{t-1} + \varepsilon_{1t} \tag{6}$$

$\Delta$ where $\Delta$ symbolizes the operator of the first difference, $\varepsilon_{1t}$ symbolises the error term, and $\omega ECT_{t-1}$ is the lagged ECT. The causal linkage amongst the tested variables in the short-run level is tested applying the Wald test's (F)statistics. Fig 5 shows the methodology structure of this research.

## 4. Empirical results and discussions

The findings of (ZA and PV) unit roots test shown in Tables 3 and 4; The findings show that $RE^C$, $R^I$, $FS^D$, $E^D$ variables are stationary at the first difference ($\Delta$). Hence, the selected variables have I(1) order of integration level. Thus, Eq (1) of this study is accepted as a tested model of co-integration.

The outcomes of Bootstrap ARDL testing of the co-integration approach are shown in Table 5. The findings show that $F_{Pesaran}$, $t_{dependent}$, and $F_{independent}$ values exceed Bootstrap CV. These findings provide significant evidence that there is a co-integration amongst $RE^C$, $R^I$, $FS^D$, $E^D$ variables.

The coefficients of $RE^C$, $R^I$, $FS^D$, $E^D$ in short and long-term were estimated through ARDL (Table 6). The findings demonstrate that a one % increase in real income increased renewable energy consumption by 0.81%, and 0.48 in the short run and long run respectively. These findings confirm a positive link among real income and renewable energy consumption in Turkey over the period from 1980 to 2016.

**Table 3. Findings of Zivot-Andrews test.**

| | At /level | | At /$\Delta$ | | |
|---|---|---|---|---|---|
| **Tested Variable** | $t-^{STAT}$ | **D-SB** | **Variables** | $t-^{STAT}$ | **D-SB** |
| Ln $RE^C_t$ | -1.315 | 2011 | $\Delta$Ln $RE^C_t$ | -8.665*** | 2013 |
| $\ln R^I_t$ | -2.001 | 2001 | $\Delta \ln R^I_t$ | -9.391*** | 2009 |
| $\ln FS^D_t$ | -2.989 | 1994 | $\Delta \ln FS^D_t$ | -8.401*** | 2006 |
| $\ln E^D_t$ | -1.425 | 2003 | $\Delta \ln E^D_t$ | -9.965*** | 2013 |

Note

***indicate the significance of variables at 1 percent level.

**Table 4. The Perron-Vogelsang test results.**

| Tested Variable | At /level | | Variables | At/ $\Delta$ | |
| --- | --- | --- | --- | --- | --- |
| | $t-^{STAT}$ | D-SB | | $t-^{STAT}$ | D-SB |
| Ln $RE^C_t$ | -2.010 | 2001 | $\Delta$Ln $RE^C_t$ | -7.661*** | 2008 |
| $\ln R^I_t$ | -1.443 | 1996 | $\Delta\ln R^I_t$ | -6.003*** | 1994 |
| $\ln FS^D_t$ | -1.998 | 1988 | $\ln FS^D_t$ | -8.365*** | 1998 |
| $\ln E^D_t$ | -2.315 | 2001 | $\Delta\ln E^D_t$ | -7.123*** | 212 |

***indicate the significance of variables at 1 percent Level.

Furthermore, the findings show that a one % increase in financial sector development increased Turkey's renewable energy consumption by 0.019% in the short run and 0.001% in the long run. These results confirm a positive link among financial sector development and renewable energy consumption over the period from 1980 to 2016.

On the other hand, the empirical outcomes of this study demonstrated that a one % increase in external debt led to a decrease in renewable energy consumption by 0.001% in the short run, and 0.017% in the long run. These results confirm a negative link among external debt and renewable energy consumption over the period from 1980 to 2016. The study suggested that any increase in the total debt in Turkey will affect negatively economic growth and d investment, which may affect negatively real income, which in turn, it may lead to a decrease in the level of renewable energy consumption.

The finding of ECM is presented in Table 6. They reflect the adjustment velocity amongst the short-run level and long-run level. The diagnostic tests finding are shown in Table 6. The $D^{t1}$ *test* finding demonstrates that P-value in this test exceed the (5%) sig level, this finding confirms that the tested model of this paper has a normal distribution. The findings of the Brush-Pagan Godfrey heteroscedasticity $D^{t2}$, the ARCH test $D^{t3}$ and the LM test $D^{t4}$ affirm that the model of this study is homoscedastic, and the tested model is well specified and homoscedastic, and there is no problem of heteroscedasticity exists in the model. The multicollinearity test $D^{t6}$ affirm that there is no multicollinearity problem in the examined model. Besides, Fig 6 (the CUSUM and CUSUM$^{Squares}$) affirm that the tested model of this research is corrected over the tested period.

It is highly essential to analyze the direction of causality among the estimated variables. Using this motivation, this study uses VECM to test both the short and long-run causal relationships that can only be used to the cointegrated series. The causal relationship amongst the tested variables helps us in crafting some appropriate policies to control oil consumption for sustainable growth in Turkey. The outcome reflects a causal interaction from real income, financial sector development, and external debt to Turkey's renewable energy consumption in the long run. Furthermore, the findings from Table 7 display that there is a unidirectional causal relationship between real income and renewable energy consumption ($\ln R^I \rightarrow \ln REC^c$).

**Table 5. The findings of the Bootstrap ARDL testing approach.**

| ARDL(0,1,1,0) | | $F_{Pesaran}$ | $t_{dependent}$ | $F_{independent}$ |
| --- | --- | --- | --- | --- |
| $(RE^C, R^I, FS^D, E^D)$ | | 6.80** | -4.45** | 5.58** |
| | | | | |
| Bootstrap-based table CV | 5% | 3.01 | -3.76 | 4.50 |

** statistical sign at 5 percent level.

**Table 6. Short and Long-run coefficients, ARDL.**

| Variable | ARDL | T-stat | p-value |
|---|---|---|---|
| $\Delta \ln R^I_t$ | 0.812*** | 2.411 | 0.00 |
| $\Delta \ln FS^D_t$ | 0.019** | 0.198 | 0.03 |
| $\Delta \ln E^D_t$ | -0.001* | -1.901 | 0.06 |
| $\ln R^I_t$ | 0.480*** | 1.651 | 0.00 |
| $\ln FS^D_t$ | 0.001* | 0.094 | 0.08 |
| $\ln E^D_t$ | -0.017* | -1.108 | 0.09 |
| $ECT_{t-1}$ | -0.620 | -3.109 | 0.00 |
| $R^2$ | 0.95 | 0.97 | 0.98 |
| Diagnostic tests | p-value | | |
| $D^{t1}$ | 0.410(0.820) | $D^{t6}$ multicollinearity test VIF | |
| $D^{t2}$ | 1.991(0.680) | Ln $RE^C_t$ 2.12 | |
| $D^{t3}$ | 1.810(0.132) | $\ln R^I_t$ 1.02 | |
| $D^{t4}$ | 1.001(0.450) | $\ln FS^D_t$ 3.42 | |
| $D^{t5}$ | 1.991(0.890) | $\ln E^D_t$ 2.13 | |

Note

*, **, ***means significance of the tested variables at 10%, 5%, 1% level, respectively.

This result confirmed that the conservation hypothesis (CH) is valid in Turkey, this hypothesis signifies the existence of a unidirectional linkage between consumption of energy and real income. Thus, an increase in real income led to an increase in the level of renewable energy consumption. This outcome is consistent with [24]. On the other hand, the outcomes show unidirectional causal linkage from financial sector development to renewable energy consumption ($\ln FS^D \rightarrow \ln RE^C$). This finding in agreement with [33] and [34]. However, this finding affirms that financial sector development can play a crucial role in attaining economic stability through credit provided to banks and markets. Hence, the outcome suggests that policymakers in Turkey must use the growth of the financial sector to minimize environmental degradation by promoting investment in energy and production through renewable energy sources. The findings of this paper are essential for Turkey in terms of diversifying the energy sources by increasing the consumption and investment in renewable energy sources. Therefore, Turkey needs to support the investment in renewable energy in the markets in order to reduce CO2 emissions in Turkey.

On the other hand, the outcomes show unidirectional causal linkage from external debt to renewable energy consumption ($\ln E^D \rightarrow \ln RE^C$), and there is the unidirectional causal linkage

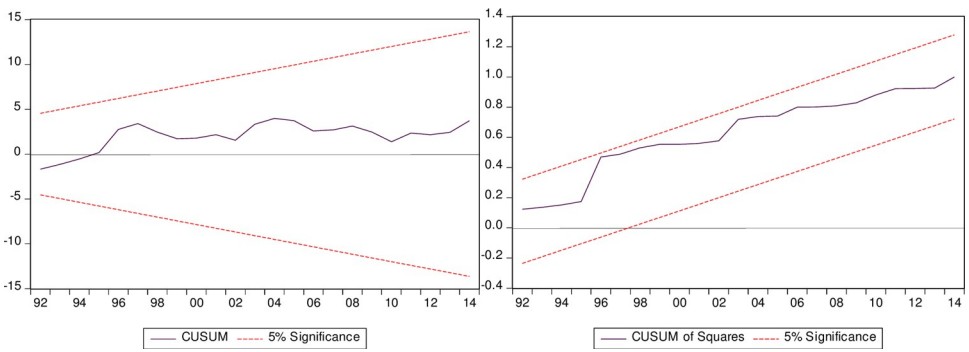

**Fig 6. Cusum and Cusum<sup>Squares</sup> tests.**

**Table 7. Results of the Granger causality testing approach.**

|  | $\Delta\ln RE^c$ | $\Delta\ln R^I$ | $\Delta\ln FS^D$ | $\Delta E^D$ | $ECT_{t-1}$ |
|---|---|---|---|---|---|
| $\Delta\ln RE^c$ | - | 6.1448* | 6.2195* | 7.6794** | -0.310** |
| $\Delta\ln R^I$ | 1.6309 | - | 6.3151* | 6.4383* | 0.521 |
| $\Delta\ln FS^D$ | 2.0974 | 4.1202 | - | 2.1231 | -0.410 |
| $\Delta E^D$ | 0.8265 | 8.5891** | 4.0564 | - | -0.329 |

Note

*, ** denote significance at 1, and 5% level.

form $\ln E^D$ to $\ln R^I$. This finding confirms that the external debts affect renewable energy consumption in Turkey through the real income channel. These results are in line with the theory of debt overhang theory. In this way, the local and foreign investments will be decreased when the total debt exceeds the repayment ability of the country, which will subsequently cause the level of productivity and income to decrease. However, the external debt affects negatively the financial strength of investment, which in turn it will affect negatively the level of energy consumption. The finding is in line with [34] and [40] who tested the linkage among public debt on the consumption of renewable energy, and confirmed that there is a positive linkage between the level of debt and renewable energy consumption.

According to the findings of the study, the external debt in Turkey is a significant barrier preventing sustainable energy development. Hence, the government policymakers should pay close attention to the negative impact of external debt on renewable energy by reducing the level of external debt. In this sense, the research suggested that government policymakers should reformulate the external debt policy to reduce the negative influence of external debt on sustainable energy development. This could potentially be achieved by removing any restrictions on international capital flow or barriers (such tariffs) on foreign capital and foreign investment. Furthermore, the policymakers in Turkey should adopt green energy policies by increasing the investment and production in renewable energy. Which in turn, it will lead to improve energy efficiency and decrease energy costs to achieve sustainable energy development.

## 5. Conclusion

In the review of literature, many empirical studies are focusing on the linkage between macro-economic factors and energy consumption. They employ different models with different selected variables and tested periods, as well as different countries. This study fills gaps in empirical literature in two different ways, the first way; limited studies that tested the linkage among external debt and renewable energy consumption. The second way; this research is the first to test the linkage among tested variables using Bootstrap ARDL. The new test of the Autoregressive Distributed lag (ARDL) test as suggested by [5] is desired over traditional co-integration tests due to its ability to predict when resolving power and size limitation issues, and its corresponding features, which have not been addressed by traditional co-integration tests. In this regard, the primary aim of this paper is to provide fresh evidence by testing the linkage between renewable energy consumption, financial development, and external debts in Turkey, using the Bootstrap ARDL test [5].

The ARDL testing approach is utilized to investigate the coefficients between the tested variables. For the direction of causality, the Granger causality approach is employed as an estimation technique. The findings from the ARDL estimations show a positive association between real income and renewable energy consumption in Turkey. Moreover, the empirical outcomes

confirm a positive linkage among financial sector development and renewable energy consumption. As a consequence, any increase in credits from banks to markets, will lead to a rise in projects and investment and strengthen risk management systems, which may lead to affect economic growth and consumption of energy. Moreover, the findings of current research show that there is a negative linkage among external debt and the level of renewable energy consumption in Turkey. The study suggests that there are several ways in which external debt can affect the consumption of energy. The first way is through the negative linkage among debt and real income when the total debt exceeds the repayment ability of the country, which will subsequently cause the level of productivity and income to decrease. However, the external debt affects negatively the financial strength of investment, which in turn it will affect negatively the level of energy consumption. The second way is through a spillover influence of external debt on the fiscal policies. An example, an increase in the levels of debts may lead to adopting policies such as (increase the tax rates) to meet the country obligations. In this sense, the high tax rates can discourage the level of saving, investment, and innovation, which in turn will affect negatively energy investment and consumption. The third way is through spillover influence of external debt on borrowing costs for the individual and markets. An increase in the borrowing cost will subsequently cause the investment to decrease; it will subsequently cause renewable energy consumption to decline. Hence, policymakers should pay close attention to the negative impact of external debt on renewable energy by reducing the level of external debt. In this sense, the research suggested that Turkey's policy-makers should reformulate the external debt policy to reduce the negative influence of external debt on sustainable energy development. This could potentially be achieved by removing any restrictions on international capital flow or barriers (such tariffs) on foreign capital and foreign investment. Furthermore, the policymakers in Turkey should adopt green energy policies by increasing the investment and production in renewable energy. Which in turn, it will lead to improve energy efficiency and decrease energy costs to achieve sustainable energy development. Furthermore, the outcome suggests that policymakers in Turkey must use the growth of the financial sector to minimize environmental degradation by promoting investment in energy and production through renewable energy sources. Hence, the findings of this paper provide valuable conclusions and recommendations for Turkey heading to sustainable and green economic growth by diversifying their energy formulate by shifting towards more investment and consumption in renewable energy sources. However, the current study provided fresh evidence by testing the linkage between renewable energy consumption, financial development, and external debts in Turkey, using the bootstrap ARDL model, over the period from 1980 to 2016. The limitations of the selected data can be attributed to that some data is not available after 2016. Future empirical studies should be devoted to the investigation of the long-term linkage between different sectors of the economy and renewable energy using different panel methods.

## Author Contributions

**Conceptualization:** Ahmed Samour.

**Data curation:** Ahmed Samour.

**Formal analysis:** Ahmed Samour.

**Funding acquisition:** Ahmed Samour.

**Investigation:** Ahmed Samour.

**Methodology:** Majdi Saleem Jabari, Ahmed Samour.

**Project administration:** Mehmet Aga.

**Software:** Ahmed Samour.

**Supervision:** Ahmed Samour.

**Validation:** Majdi Saleem Jabari, Ahmed Samour.

**Visualization:** Ahmed Samour.

**Writing – original draft:** Ahmed Samour.

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
