## [Decision Letter · Decision Letter 0]

23 Dec 2021

PONE-D-21-36442The linkage between renewable energy consumption and financial sector development in Turkey: The role of external debtPLOS ONE

Dear Dr. Samour,

Thank you for submitting your manuscript to PLOS ONE. After careful consideration, we feel that it has merit but does not fully meet PLOS ONE’s publication criteria as it currently stands. Therefore, we invite you to submit a revised version of the manuscript that addresses the points raised during the review process.

The manuscript requires further revisions regarding research contribution, prior literature extension, econometric outcomes’ refinements, policy and practical implications, along with English language improvement.

We look forward to receiving your revised manuscript.

Kind regards,

Stefan Cristian Gherghina, PhD. Habil.

Academic Editor

PLOS ONE

Journal Requirements:

"The funders had no role in study design, data collection and analysis, decision to publish, or preparation of the manuscript"

Reviewers' comments:

Reviewer's Responses to Questions

**Comments to the Author**

1. Is the manuscript technically sound, and do the data support the conclusions?

Reviewer #1: Yes

Reviewer #2: Yes

Reviewer #3: Yes

2. Has the statistical analysis been performed appropriately and rigorously? 

Reviewer #1: Yes

Reviewer #2: Yes

Reviewer #3: Yes

3. Have the authors made all data underlying the findings in their manuscript fully available?

Reviewer #1: Yes

Reviewer #2: Yes

Reviewer #3: Yes

4. Is the manuscript presented in an intelligible fashion and written in standard English?

Reviewer #1: Yes

Reviewer #2: Yes

Reviewer #3: Yes

5. Review Comments to the Author

Reviewer #1: I am pleased to review the manuscript titled “.The linkage between renewable energy consumption and financial sector development in Turkey: The role of external debt” The authors have done well in structuring their study. Overall, the sections are elaborately written. Hence, I believe that this study can be considered for publication provided the authors are ready to revise their manuscript as per the minor comments provided below:

1. Title: If it is possible the authors may attempt to shorten the title.

2. Abstract: The abstract should include 1/2 lines at the beginning to provide a brief background of the study.

3. Introduction: The contributions of the study should be precisely highlighted.

4. The authors can provide more recent references related with renewable energy promotion.

Doğan, B., Driha, O. M., Balsalobre Lorente, D., & Shahzad, U. (2021). The mitigating effects of economic complexity and renewable energy on carbon emissions in developed countries. Sustainable Development, 29(1), 1-12.

Balsalobre-Lorente, D., & Leitão, N. C. (2020). The role of tourism, trade, renewable energy use and carbon dioxide emissions on economic growth: evidence of tourism-led growth hypothesis in EU-28. Environmental Science and Pollution Research, 27(36), 45883-45896.

5. Conclusion: Please mention the limitations of the study and mention the future research direction as well.

6. Please proof read the manuscript before submitting the revision.

Reviewer #2: In this paper, Bootstrap ARDL test is used to investigate the long-term and short-term linkages between renewable energy consumption and real income, financial development, as well as external debts in Turkey. For the direction of causality, the Granger causality is used as the estimation technique. Diagnostic tests (Normality test, BP test, ARCH test, LM test and Ramsey-Reset test) prove that the model in this paper is well set. And the CUSUM and CUSUM Squares tests affirm that the tested model of this research is corrected over the tested period. the findings of this paper provide valuable conclusions and recommendations for Turkey heading to sustainable and green financial sector. The contributions of this paper can be summarized as follows:

1. Enrich relevant literatures on the impact of real income, financial development and external debts on renewable energy consumption.

2. The use of newly ARDL test (Bootstrap ARDL) provides fresh evidence for studies on the impact of tested variables on renewable energy consumption in Turkey.

For further improving the paper, I have a number of comments and suggestions:

1. Equation 3 and Equation 4 are exactly the same, please rearrange this part to make it more organized.

2. Lines 65-68, you mentioned “the optimal of lags”. Please give the selection basis of optimal lags in the result analysis, is it AIC? SC? Or some other criterion?

3. In details. First, in lines 80-81, ARCH test should be heteroscedasticity test. Second, the inconsistent references. For example, you mention in section 2.1 that [21] supports the FH hypothesis, while in lines 178-183 you said that "This result confirmed that the conservation hypothesis (CH) is valid in Turkey" and that "This outcome is consistent with ([20]; [21])". Lines 207 -210 have the same problem for [32] and [36].

4. Many scholars have proved the correlation between financial development and real income ([25]; [26]; [27]), and between external debt and real income ([34]; [14]; [33]). Then, is there a high degree of collinearity when these three variables are included in the model? I suggest a multicollinearity test.

Reviewer #3: The author seeks to investigate the linkage among renewable energy consumption, financial development, and external debts in Turkey. Author could improve the manuscript based on the suggestions below;

1. on page 2, the author mentioned that “…limited studies tested the possible influence of external debt and financial development on their frameworks.”. This is not entirely true. There are several studies on financial development on renewable energy. see the study in this link https://doi.org/10.1016/j.apenergy.2021.118023 . You can also find several studies in the reference of the paper in the link above.

Please provide strong justification and contributions of your study. please the contributions on page 4 should be moved above and strengthened.

2. the current structure of the study seeks to investigate correlation between the variables of interest, given the potential endogeneity of the financial development, external debt and real income. I suggest author to desist from the use of “impact” or “effects” in the study.

3. on page 9, the whole of the second paragraph which starts with the sentence “….To explore the linkage ….” needs to be rewritten. For example, the sentence that starts with “..However, the new technique contains …” is not clear.

4. given that there could be cofounders which may drive the results, author should highlight the limitations of the study, especially the method.

6. PLOS authors have the option to publish the peer review history of their article (what does this mean?). If published, this will include your full peer review and any attached files.

Reviewer #1: No

Reviewer #2: No

Reviewer #3: No

---

## [Author Response · Author response to Decision Letter 0]

10 Feb 2022

Response to Reviewers ‘comments

Reviewer: 1 

Comment: 

Title: If it is possible the authors may attempt to shorten the title.

Response: Thank you very much for your valuable comment; we have changed the title to “Financial sector development, external debt, and Turkey’s renewable energy consumption”

Comment: 

2. Abstract: The abstract should include 1/2 lines at the beginning to provide a brief background of the study.

Response: Thank you very much for your valuable comment; we have added 2 lines at the beginning of the Abstract. 

Comment: 

Introduction: The contributions of the study should be precisely highlighted.

Response: Thank you very much for your valuable comment; we have revised the contributions of the study in the introduction section. 

Comment: 

The authors can provide more recent references related with renewable energy promotion.

Doğan, B., Driha, O. M., Balsalobre Lorente, D., & Shahzad, U. (2021). The mitigating effects of economic complexity and renewable energy on carbon emissions in developed countries. Sustainable Development, 29(1), 1-12.

Balsalobre-Lorente, D., & Leitão, N. C. (2020). The role of tourism, trade, renewable energy use and carbon dioxide emissions on economic growth: evidence of tourism-led growth hypothesis in EU-28. Environmental Science and Pollution Research, 27(36), 45883-45896.

Response: Thank you very much for your valuable comment; we have added some recent studies related to renewable energy promotion for example Doğan, B., Driha, O. M., Balsalobre Lorente, D., & Shahzad, U. (2021). The mitigating effects of economic complexity and renewable energy on carbon emissions in developed countries. Sustainable Development, 29(1), 1-12.

Balsalobre-Lorente, D., & Leitão, N. C. (2020). The role of tourism, trade, renewable energy use and carbon dioxide emissions on economic growth: evidence of tourism-led growth hypothesis in EU-28. Environmental Science and Pollution Research, 27(36), 45883-45896.

Comment: 

Conclusion: Please mention the limitations of the study and mention the future research direction as well.

Response: Thank you very much for your valuable comment; we have added limitations of the study and future research direction in the conclusion section. 

Comment: 

Please proofread the manuscript before submitting the revision.

Response: Thank you very much for your valuable comment, we have carefully proofread the manuscript revised, we checked all typos, grammar, and spelling in our manuscript.

Finally, we would like to thank you for your valuable suggestions. We believe your suggestions improved our study.

Reviewer 2

Comments 

In this paper, the Bootstrap ARDL test is used to investigate the long-term and short-term linkages between renewable energy consumption and real income, financial development, as well as external debts in Turkey. For the direction of causality, the Granger causality is used as the estimation technique. Diagnostic tests (Normality test, BP test, ARCH test, LM test and Ramsey-Reset test) prove that the model in this paper is well set. And the CUSUM and CUSUM Squares tests affirm that the tested model of this research is corrected over the tested period. the findings of this paper provide valuable conclusions and recommendations for Turkey heading to sustainable and green financial sector. The contributions of this paper can be summarized as follows:

1. Enrich relevant literatures on the impact of real income, financial development and external debts on renewable energy consumption.

In this paper, Bootstrap ARDL test is used to investigate the long-term and short-term linkages between renewable energy consumption and real income, financial development, as well as external debts in Turkey. For the direction of causality, the Granger causality is used as the estimation technique. Diagnostic tests (Normality test, BP test, ARCH test, LM test and Ramsey-Reset test) prove that the model in this paper is well set. And the CUSUM and CUSUM Squares tests affirm that the tested model of this research is corrected over the tested period. the findings of this paper provide valuable conclusions and recommendations for Turkey heading to sustainable and green financial sector. The contributions of this paper can be summarized as follows:

1. Enrich relevant literatures on the impact of real income, financial development and external debts on renewable energy consumption.

2. The use of newly ARDL test (Bootstrap ARDL) provides fresh evidence for studies on the impact of tested variables on renewable energy consumption in Turkey.

For further improving the paper, I have a number of comments and suggestions:

1. Equation 3 and Equation 4 are exactly the same; please rearrange this part to make it more organized.

 Response: Thank you very much for your comments. We have revised the equations 2, and 3 based on your comments. 

Comment: 

2. Lines 65-68, you mentioned “the optimal of lags”. Please give the selection basis of optimal lags in the result analysis, is it AIC? SC? Or some other criterion?

Response: Thank you very much for your comments and suggestion. We have revised it. 

Comment: 

3. In details. First, in lines 80-81, ARCH test should be heteroscedasticity test. Second, the inconsistent references. For example, you mention in section 2.1 that [21] supports the FH hypothesis, while in lines 178-183 you said that "This result confirmed that the conservation hypothesis (CH) is valid in Turkey" and that "This outcome is consistent with ([20]; [21])". Lines 207 -210 have the same problem for [32] and [36].

Response: Thank you very much for your valuable suggestion; we have revised that ARCH test.

For the references: we have revised and corrected it. 

Comment: 

Many scholars have proved the correlation between financial development and real income ([25]; [26]; [27]), and between external debt and real income ([34]; [14]; [33]). Then, is there a high degree of collinearity when these three variables are included in the model? I suggest a multicollinearity test.

Response: Thank you very much for your valuable comment; we have used the multicollinearity test to affirm that there is no multicollinearity problem in the examined model.

Finally, we would like to thank you for your valuable comments and suggestions. We believe your suggestions improved our study.

Reviewer 3

Comment: 

Reviewer #3: The author seeks to investigate the linkage between renewable energy consumption, financial development, and external debts in Turkey. Author could improve the manuscript based on the suggestions below;

1. on page 2, the author mentioned that “…limited studies tested the possible influence of external debt and financial development on their frameworks.”. This is not entirely true. There are several studies on financial development on renewable energy. see the study in this link https://doi.org/10.1016/j.apenergy.2021.118023 . You can also find several studies in the reference of the paper in the link above.

Please provide strong justification and contributions of your study. Please the contributions on page 4 should be moved above and strengthened.

Response: Thank you very much for your valuable comment; we have revised and deleted the financial development form this paragraph. We have updated the contributions of our study and moved above.

Comment: 

The current structure of the study seeks to investigate correlation between the variables of interest, given the potential endogeneity of the financial development, external debt and real income. I suggest author to desist from the use of “impact” or “effects” in the study.

Response: Thank you very much for your valuable comment; we have revised based on your suggestion

Comment: 

on page 9, the whole of the second paragraph which starts with the sentence “….To explore the linkage ….” needs to be rewritten. For example, the sentence that starts with “..However, the new technique contains …” is not clear.

Response: Thank you very much for your valuable comment; we have revised it. 

Comment: 

given that there could be cofounders which may drive the results, author should highlight the limitations of the study, especially the method.

Response: Thank you very much for your valuable comment; we have added the limitations of the study, and future studies at conclusion section.

---

## [Decision Letter · Decision Letter 1]

7 Mar 2022

Financial sector development, external debt, and Turkey’s renewable energy consumption

PONE-D-21-36442R1

Dear Dr. Samour,

We’re pleased to inform you that your manuscript has been judged scientifically suitable for publication and will be formally accepted for publication once it meets all outstanding technical requirements. In this regard, the author(s) should implement the remaining revisions as recommended by the second referee.

Kind regards,

Stefan Cristian Gherghina, PhD. Habil.

Academic Editor

PLOS ONE

Additional Editor Comments (optional):

Reviewers' comments:

Reviewer's Responses to Questions

**Comments to the Author**

1. If the authors have adequately addressed your comments raised in a previous round of review and you feel that this manuscript is now acceptable for publication, you may indicate that here to bypass the “Comments to the Author” section, enter your conflict of interest statement in the “Confidential to Editor” section, and submit your "Accept" recommendation.

Reviewer #1: All comments have been addressed

Reviewer #2: All comments have been addressed

Reviewer #3: All comments have been addressed

2. Is the manuscript technically sound, and do the data support the conclusions?

Reviewer #1: Yes

Reviewer #2: Yes

Reviewer #3: Yes

3. Has the statistical analysis been performed appropriately and rigorously? 

Reviewer #1: Yes

Reviewer #2: Yes

Reviewer #3: Yes

4. Have the authors made all data underlying the findings in their manuscript fully available?

Reviewer #1: (No Response)

Reviewer #2: Yes

Reviewer #3: (No Response)

5. Is the manuscript presented in an intelligible fashion and written in standard English?

Reviewer #1: Yes

Reviewer #2: Yes

Reviewer #3: (No Response)

6. Review Comments to the Author

Reviewer #1: (No Response)

Reviewer #2: This paper does an empirical analysis on the nexus of external debt, financial development, real income and renewable energy consumption in Turkey using bootstrap ARDL testing, it can enrich literatures about relationship between financial development and renewable energy consumption. It has been revised appropriately according to the comments and suggestions raised by reviewers. Some mistakes have been corrected, some latest related literatures have been considered and updated in this paper and some typos and sentences have also been revised.

For further improving the paper, I have two more comments and suggestions:

Is it suitable for the last keywords of “Energy” as it is mainly analyzing about the nexus of renewable energy consumption, financial development and external debt rather than energy in this paper? May it be more appropriate to change the last keywords to “renewable energy consumption”?

In line 320, a more “Δ” appears before the word “where”, please double check it.

Reviewer #3: (No Response)

7. PLOS authors have the option to publish the peer review history of their article (what does this mean?). If published, this will include your full peer review and any attached files.

Reviewer #1: **Yes: **Daniel Balsalobre

Reviewer #2: No

Reviewer #3: No

---

## [Editor Report · Acceptance letter]

21 Apr 2022

PONE-D-21-36442R1 

Financial sector development, external debt, and Turkey’s renewable energy consumption 

Dear Dr. Samour:

I'm pleased to inform you that your manuscript has been deemed suitable for publication in PLOS ONE. Congratulations! Your manuscript is now with our production department. 

Kind regards, 

on behalf of

Dr. Stefan Cristian Gherghina 

Academic Editor

PLOS ONE